# Evaluation of Blood Intercellular Adhesion Molecule-1 (ICAM-1) Level in Obstructive Sleep Apnea: A Systematic Review and Meta-Analysis

**DOI:** 10.3390/medicina58101499

**Published:** 2022-10-21

**Authors:** Mohammad Moslem Imani, Masoud Sadeghi, Mohammad Amir Gholamipour, Annette Beatrix Brühl, Dena Sadeghi-Bahmani, Serge Brand

**Affiliations:** 1Department of Orthodontics, Kermanshah University of Medical Sciences, Kermanshah 67158-47141, Iran; 2Department of Biology, Science and Research Branch, Islamic Azad University, Tehran 14778-93855, Iran; 3Students Research Committee, Kermanshah University of Medical Sciences, Kermanshah 67158-47141, Iran; 4Center for Affective, Stress and Sleep Disorders (ZASS), Psychiatric University Hospital Basel, 4002 Basel, Switzerland; 5Sleep Disorders Research Center, Kermanshah University of Medical Sciences, Kermanshah 67158-47141, Iran; 6Department of Psychology, Stanford University, Stanford, CA 94305, USA; 7Department of Sport, Exercise and Health, Division of Sport Science and Psychosocial Health, University of Basel, 4052 Basel, Switzerland; 8Substance Abuse Prevention Research Center, Kermanshah University of Medical Sciences, Kermanshah 67198-51115, Iran; 9School of Medicine, Tehran University of Medical Sciences, Tehran 14166-34793, Iran

**Keywords:** obstructive sleep apnea, blood, intercellular adhesion molecule-1, meta-analysis

## Abstract

*Background and objective*: Intercellular adhesion molecule-1 (ICAM-1) appears to be an active and important biomarker for decreasing the risk of cardiovascular issues among individuals with obstructive sleep apnea (OSA). Herein, a systematic review and meta-analysis was designed to probe whether plasma/serum ICAM-1levels are different in adults with OSA compared to adults with no OSA, as well as adults with severe OSA compared to adults with mild/moderate OSA. *Materials and methods*: A thorough and systematic literature search was performed in four databases (PubMed/Medline, Web of Science, Scopus, and Cochrane Library) until 17 July 2022, without any age and sample size restrictions to retrieve the relevant articles. The standardized mean difference (SMD) along with a 95% confidence interval (CI) of plasma/serum of ICAM-1 levels was reported. Analyses, including sensitivity analysis, subgroup analysis, trial sequential analysis, meta-regression, and a funnel plot analysis, were performed in the pooled analysis. *Results*: A total of 414 records were identified in the databases, and 17 articles including 22 studies were entered into the meta-analysis. The pooled SMD of serum/plasma ICAM-1 levels in adults with OSA compared to controls was 2.00 (95%CI: 1.41, 2.59; *p* < 0.00001). The pooled SMD of serum/plasma ICAM-1 levels in adults with severe compared to mild/moderate OSA was 3.62 (95%CI: 1.74, 5.51; *p* = 0.0002). Higher serum/plasma ICAM-1 levels were associated with a higher mean age of controls, higher scores for the apnea-hypopnea index, and with a lower mean age of adults with OSA and with smaller sample sizes. *Conclusions*: Th results of the present meta-analysis showed that serum/plasma ICAM-1 levels in adults with OSA was higher than serum/plasma ICAM-1 levels in controls. Similarly, serum/plasma ICAM-1 levels in adults with severe OSA were higher compared to serum/plasma ICAM-1 levels of adults with mild or moderate OSA. Therefore, ICAM-1 may be used as an additional diagnostic and therapeutic biomarker in adults with OSA.

## 1. Introduction

Obstructive sleep apnea (OSA) is the most prevalent sleep-related breathing disorder [1]. OSA is associated with daytime sleepiness, poor concentration, short attention, increased fatigue, and higher symptoms of depression [2,3]. For diagnosis and definition of OSA, the apnea-hypopnea index (AHI = number of (apneas + hypopneas)/total sleep time) is reported. For adults, cut-off values are: AHI  <  5 events/h: normal or no OSA; 5 ≤ AHI  <  15 events/h: mild OSA; 15  ≤  AHI  <  30 events/h: moderate OSA; AHI  ≥  30 events/h: severe OSA [4]. The main risk factors of OSA are older age, male gender, higher obesity, and the occurrence of craniofacial/upper airway abnormalities [5,6]. In 2019, a literature-based worldwide analysis [7] estimated that 936 million adults aged 30–69 years (male and female) suffer from mild to severe OSA, and 425 million adults aged 30–69 years have moderate to severe OSA. Further, the number of individuals with OSA was highest in China, followed by the USA, Brazil, and India [7].

OSA has an evidence-based independent relationship with cardiovascular disease, metabolic syndrome, hypertension [8,9,10], asthma [11], stroke [12], and neurological disorder [13]. In addition to environmental factors, OSA appears to be also a complex genetic illness likely caused by several genetic factors [14]. In the same vein, several recent meta-analyses reported the association between the risk of OSA and specific polymorphisms, such as interleukin (IL)-6, leptin receptor, and matrix metalloproteinase-9 [15,16,17], blood levels of biomarkers such as adiponectin, C-reactive protein, IL-6, and cortisol [18,19,20,21], smoking [22,23], along with greater alcohol and caffeine consumption [23].

Intercellular adhesion molecule-1 (ICAM-1) is a transmembrane glycoprotein that is overexpressed in many pathological states [24]. ICAM-1 is upregulated in the event of inflamed tissue, and ICAM-1 functions as an adhesion receptor on endothelial and epithelial cells. As such, ICAM-1 is a critical receptor that may facilitate the initiation and progression of inflammatory responses [25,26]. Therefore, ICAM-1 can be an important marker of inflammation [24]. This marker, as a cell surface glycoprotein and an adhesion receptor, can regulate the recruitment of leukocytes from the circulation to inflammation sites [27]. ICAM-1 expression can be strongly induced in vascular endothelial, epithelial, and immune cells in response to stimulation of inflammatory [27]. For the following reasons, ICAM-1 is of particular interest: ICAM-1 has an important role in the functioning of the blood-brain barrier, and as such, ICAM-1 plays a significant role in the biology of the emergence and maintenance of psychiatric disorders. Further, among individuals with OSA and undergoing continuous positive airway pressure (CPAP) therapy, outcome success was associated with decreased ICAM-1 level and with lower odds of cardiovascular issues [28]. Next, while several studies [29,30,31] have reported associations between ICAM-1 level and the risk of OSA, results were mixed, in that either no association was observed, or individuals with OSA had either increased or decreased levels of ICAM-1 when compared to the general population. Contrary to this, a meta-analysis published in 2013 and consisting of eight articles (12 studies) [32] reported higher blood levels of ICAM-1 in adults with OSA compared to controls.

To either confirm or reject previous results, we performed a systematic review and meta-analysis including more case-control studies, and we reported the association of level of ICAM-1 with the severity of OSA. In addition, unlike a previous meta-analysis [32], we did not include studies with less than ten cases and without reporting AHI scores. The reason for adopting this procedure is that the results should turn out to be more robust. In addition, we added the following analyses: trial sequential analysis (TSA); radial and L’Abbe plots; trim-and-fill method; sensitivity and subgroup analyses; such additional analyses should help to improve the overall quality of the present findings. Therefore, the aim of the meta-analysis was to ascertain whether plasma/serum ICAM-1 levels are different in adults with OSA compared to adults with no OSA, as well as adults with severe OSA compared to adults with mild/moderate OSA.

## 2. Materials and Methods

### 2.1. Study Design

We reported the systematic review according to the PRISMA guideline [33]. The two PECO (Population, Exposure, Comparator, and Outcome) questions [34,35] were: 1. Are plasma/serum ICAM-1 levels different in adults with OSA compared to adults with no OSA? 2. Are plasma/serum ICAM-1levels different in adults with severe OSA compared to adults with mild/moderate OSA? (P: human cases with and without OSA, E: OSA disease, C: adults with OSA compared to controls and/or adults with mild/moderate compared to severe OSA; O: and the plasma/serum ICAM-1 level).

### 2.2. Identification of Articles

A search was comprehensively performed by one author (M.S) in the databases of PubMed/Medline, Web of Science, Scopus, and Cochrane Library until 17 July 2022, with age and sample size restrictions to retrieve the relevant articles. The search strategy was: (“obstructive sleep apnea” or “sleep apnea” or “OSA” or “obstructive sleep apnea-hypopnea syndrome” or “OSAHS” or “obstructive sleep apnea syndrome” or “OSAS”) and (“inter-cellular adhesion molecule *” or “intercellular adhesion molecule *” or “ICAM” or “cell adhesion molecule” or “CAM”), (see the search strategy in PubMed in Appendix A). Moreover, the citations of the retrieved reviews and meta-analyses in relation to the subject were tested to ensure that no study was missed, and the same author (M.S) evaluated their titles/abstracts. Then, the full-text articles meeting the criteria were downloaded. Another author (M.M.I) re-evaluated the process. A disagreement between two previous authors was resolved by a third author (S.B).

### 2.3. Eligibility Criteria

Inclusion criteria were: (1) any study including adults with OSA vs. controls or adults with severe vs. mild/moderate OSA aged ≥ 18 years; (2) any study including adults with OSA and controls without any medical and surgical treatments; (3) studies reporting plasma/serum ICAM-1 level in adults with OSA and controls; (4) OSA was polysomnographically diagnosed defined as AHI ≥ 5 events/h; (5) adults with OSA had no other systemic diseases, such as with rhinitis, sinusitis, respiratory infections, and systemic infections, coronary artery disease (CAD), diabetes mellitus, renal diseases, or stenosis; (6) controls had no OSA or systemic disease (see item 5); and (7) serum/plasma samples were obtained between 8 and 10 am. Exclusion criteria were: (1) meta-analyses, reviews, book chapters, and conference papers; (2) studies with incomplete data; (3) studies including control groups with AHI ≥ 5 events/h; (4) studies including pediatric samples aged < 18 years; (5) studies including adults with OSA with other diseases (see item 6) or under medical and surgical treatments; (6) studies including less than 10 individuals in one or both groups of adults with OSA or control groups).

### 2.4. Data Collection

Two authors (M.S and M.M.I) separately excerpted the data of the articles selected for the meta-analysis. Excerpted data included: the first author, the publication year, the ethnicity of individuals, ICAM-1 sampling, quality score, the sample size, number of adults with OSA and controls, OSA type, the mean of plasma/serum of ICAM-1 level in the groups; and mean body mass index (BMI), age, and AHI of the groups.

### 2.5. Quality Evaluation

The quality evaluation of the studies was performed by one author (M.S) using the Newcastle-Ottawa scale (NOS) [36], according to which the number 9 was considered as the maximum score/point.

### 2.6. Statistical Analysis

The standardized mean difference (SMD) along with a 95% confidence interval (CI) of plasma/serum of ICAM-1 level between adults with OSA and controls and adults with severe and mild/moderate OSA were extracted by the Review Manager software version 5.3 (RevMan 5.3; the Cochrane Collaboration, the Nordic Cochrane Centre, Copenhagen, Denmark). A *p*-value (2-sided) < 0.05 was considered statistically significant. A *P*_heterogeneity_ < 0.1 (I^2^ > 50%) recognized significant heterogeneity and therefore a random-effects model [37] and, differently, a fixed-effect model [38] were utilized.

In a number of studies, the data were extracted from a graph utilizing GetData Graph Digitizer software version 2.26.0.20 (GetData Pty Ltd., Kogarah, Australia).

The subgroup analysis and the fixed-effect meta-regression analysis (univariate analysis) were done according to several variables. To evaluate the stability or consistency of pooled results, both “one-study-removed” and “cumulative” analyses as two sensitivity analyses were used.

The degree of publication bias was determined using a funnel plot and Egger’s regression test [39] to assess possible publication bias in a meta-analysis through funnel plot asymmetry. The potential publication bias was tested by Begg’s test [40]. The *p*-values of Egger’s and Begg’s tests were extracted and a *p*-value (2-sided) < 0.10 introduced the being of the publication bias. The Comprehensive Meta-Analysis software version 2.0 (CMA 2.0; Biostat Inc., Englewood, NJ, USA) was used to analyze publication bias, meta-regression, and sensitivity analyses.

To address false-negative or false-positive results from meta-analyses [41], TSA was accomplished using TSA software (version 0.9.5.10 beta) (Copenhagen Trial Unit, Centre for Clinical Intervention Research, Rigshospitalet, Copenhagen, Denmark) [42]. The required information size (RIS) was computed (alpha risk = 5% and beta risk = 20%) that D^2^ was 100% for plasma/serum ICAM-1 levels. The mean difference (MD) and variance were based on empirical assumptions constructed automatically by the software. If the Z-curve interrupted the RIS line or the borderline or arrived at the futility zone, enough individuals were located in the studies and the conclusion was reliable.

## 3. Results

### 3.1. Study Selection

Out of 414 records identified among the databases, after removing duplicates based on title, 263 records remained (Figure 1). Then, 207 irrelevant records were excluded based on title/abstract and 56 full-text articles were further evaluated. Of those 56 publications, 39 articles were excluded. Finally, 17 articles [29,30,31,43,44,45,46,47,48,49,50,51,52,53,54,55,56] including 22 studies were entered into the meta-analysis. 

### 3.2. Studies’ Characteristics

The seventeen articles were published from 2002 to 2022 (Table 1). Eight articles [30,44,49,51,52,53,54,56] reported data on Asians, seven [29,45,46,47,48,50,55] reported data on in Caucasians, and two studies [31,43] reported data from countries with mixed ethnicity. All studies included 1036 adults with OSA and 491 controls. One article [50] did not include a control group but was relevant for the severity of OSA (reporting severe vs. mild/moderate OSA). Thirteen articles [29,30,31,44,45,46,48,49,50,51,52,54,55] reported serum and four [43,47,53,56] plasma ICAM-1 levels. In addition, Table 1 shows other variables including mean age BMI, and AHI of two groups. The quality score with details for each study is reported in Appendix A and Table 1 shows extracted total point of each study. Among all studies, three studies had no high quality (point < 7). Concerning selection, eight out of 17 studies (47.1%) had no description of source (definition of control) and one out of 17 (5.9%) had no description (selection of control). Concerning comparability, two out of 17 studies (11.8%) did not match age between two groups (control for age) and six out of 17 (35.3%) did not match sex (control for sex) between two groups. Concerning exposure, just one out of 17 (5.9%) missed several individuals in final analysis or had rate differences and no designation (nonrsponse rate). 

### 3.3. Forest Plot Analyses

Results of random-effects forest plot analysis of serum/plasma ICAM-1 level in adults with OSA compared to controls based on twenty-one studies are reported in Table 2. The pooled SMD was 2.00 (95%CI: 1.41, 2.59; *p* < 0.00001) and I^2^ = 96% (*P*_heterogeneity_ < 0.00001). Therefore, the serum/plasma ICAM-1 level in adults with OSA was significantly higher than controls.

Results of random-effects forest plot analysis of serum/plasma ICAM-1 levels in adults with severe compared to mild/moderate OSA based on five studies are reported in Table 3. The pooled SMD was 3.62 (95%CI: 1.74, 5.51; *p* = 0.0002) and I^2^ = 98% (*P*_heterogeneity_ < 0.00001). Therefore, the serum/plasma ICAM-1 level in adults with severe OSA were significantly higher than the mild/moderate OSA. 

Forest plots of two analyses (serum/plasma ICAM-1 levels in adults with OSA compared to controls and serum/plasma ICAM-1 levels in adults with severe compared to mild/moderate OSA) are shown in Appendix A, respectively.

### 3.4. Sensitivity Analysis

Plots of one-study-remove and cumulative analyses are shown in Appendix A, respectively. The analyses showed the stability of the pooled result of serum/plasma ICAM-1 levels in adults with OSA compared to controls.

### 3.5. Subgroup Analysis

The subgroup analysis of ICAM-1 levels in adults with OSA compared to controls is identified in Table 4. The results showed that ethnicity, sample size, and blood sampling were not effective factors for the pooled results.

### 3.6. Meta-Regression

Fixed-effects meta-regression analysis of ICAM-1 levels in adults with OSA compared to controls is identified in Table 5. The findings showed that age, AHI, and sample size could be confounding factors for the pooled results. Serum/plasma ICAM-1 level increased while the mean age of controls, AHI score, and quality point increased, whereas the mean age of adults with OSA and sample size reduced.

### 3.7. Trial Sequential Analysis

The TSA of serum/plasma ICAM-1 levels in adults with OSA compared to controls is shown in Figure 2. Based on TSA result, there are sufficient cases for the pooled SMD of serum/plasma ICAM-1 levels in adults with OSA compared to controls, and therefore further studies will not need to be conducted.

### 3.8. Heterogeneity Plots

The Galbraith (or Radial) and L’Abbe plots of serum/plasma ICAM-1 levels in adults with OSA compared to controls are shown in Appendix A, respectively. The radial plot shows that 16 out of 21 studies are heterogeneous and four studies have high heterogeneity (studies 2, 7, 18, and 19). L’Abbe plot shows that most studies are heterogeneous and eleven out of 21 have high heterogeneity (studies 6–9, 12–16, 18, and 19).

### 3.9. Publication Bias

The funnel plot of serum/plasma ICAM-1 levels in adults with OSA compared to controls is shown in Appendix A reports the results of the trim-and-fill method. Egger’s (*p* = 0.0003) and Begg’s (*p* = 0.003) tests showed publication bias.

For serum/plasma ICAM-1 level and 9 imputed studies, under the fixed-effects model, the point estimate and pseudo 95% CI for the combined studies is 0.967 (0.856, 1.077). Using the trim-fill method, the imputed point estimate is 0.598 (0.498, 0.698). In addition, under the random-effects model, the point estimate and 95% CI for the combined studies is 2.043 (1.448, 2.639). Using the trim-fill method, the imputed point estimate is 0.665 (0.029, 1.301).

The overall effect sizes on serum/plasma ICAM-1 level reported in the forest plot appeared valid, with significant publication bias effect based on fixed-effects and random-effects models, because the observed estimates had a high difference from the adjusted estimates.

## 4. Discussion

Cellular adhesion molecules are good factors of endothelial dysfunction and vascular inflammation [57,58]. The ICAM-1 as a cellular adhesion molecule is a marker widely used in OSA studies to investigate the level of inflammation [47]. This marker is a cell surface receptor that involves the interaction between keratinocytes and leukocytes [59]. The main findings from the present meta-analysis were that serum/plasma ICAM-1 level in adults with OSA were higher than that in controls, and serum/plasma ICAM-1 level in adults with severe OSA were higher than serum/plasma ICAM-1 level in adults in mild/moderate OSA.

Systemic ICAM-1 has been particularly reported to play a significant role in leukocyte migration to the inflamed area [60,61]. Carpagnano et al. [47] reported that a positive correlation was across ICAM-1 level with AHI, percentage of total sleep time with oxyhemoglobin saturation < 90%, BMI, neck circumference, IL-8 levels, as well as ICAM-1 and E-selectin [43] in adults with OSA. Research [62] showed that the ICAM levels were found to have a weak relationship with BMI and abdominal circumference in adults with OSA. A correlation between ICAM-1 and AHI has been previously observed in adults with OSA [32,46,47]. Research showed the correlation between higher age and ICAM-1 level in OSA [46]. In addition, a correlation between ICAM-1 and IL-6 levels has been reported [63]. The present meta-analysis confirmed the correlation between higher serum/plasma levels of ICAM-1 and higher scores for AHI and older age, while BMI scores were unrelated to serum/plasma levels of ICAM-1.

Chang et al. [52] concluded that adults with OSA without metabolic syndrome had significantly higher carotid intima-media thickness (IMT) and higher ICAM-1 level, and besides age and mean systolic blood pressure, ICAM-1 level was associated independently with carotid IMT. Another research [43] reported that levels of ICAM-1 were significantly increased in CAD in adults with moderate-to-severe OSA, in comparison to those of a matched control group. It has been shown that ICAM-1 levels were related to an elevation in cardiovascular mortality in patients with CAD [64,65], such that ICAM-1 levels were independently associated with ischemic reactive hyperemia in adults with OSA [50]. Given this, ICAM-1 level can be considered a reliable predictor of CAD, similar to C-reactive protein (CRP) [66]. In contrast, serum ICAM-1 has traditionally been related to endothelial dysfunction linked with vascular damage [60], which is associated with OSA [30,49,53,60], and the present meta-analysis also confirmed this result. Therefore, ICAM-1 levels appear to be a reliable factor in OSA. As such, it appears conceivable that changes in ICAM-1 level might be added to the standard routine assessment of OSA.

In the present meta-analysis, there was bias due to the difference in age or sex between two groups and even the selection and definition of controls in several studies. In addition, meta-regression reported that the quality score could be an effective factor. Therefore, well-designed studies with high quality are needed in the future to be able to report a meta-analysis with better results and less or without bias.

Despite the robust pattern of results, the following limitations should be considered: (1) High heterogeneity across and between the studies. (2) Biases were most common in the papers. (3) Most studies had outliers. (4) Most studies included small sample sizes. In addition, stability of results, number of sufficient cases in the pooled analysis, and deleting the studies with less than ten cases in one or two groups were the most important strengths of the present meta-analysis.

## 5. Conclusions

The main results of the present meta-analysis were that serum/plasma ICAM-1 level in adults with OSA was higher than serum/plasma ICAM-1 level in controls, and that adults with severe OSA had higher serum/plasma ICAM-1 level than adults with mild/moderate OSA. Therefore, it is conceivable that serum/plasma ICAM-1 level might be used as a biomarker for the progression of OSA and for the therapy of OSA. Further, the present pattern of results might instigate further research in this field.

## Figures and Tables

**Figure 1 medicina-58-01499-f001:**
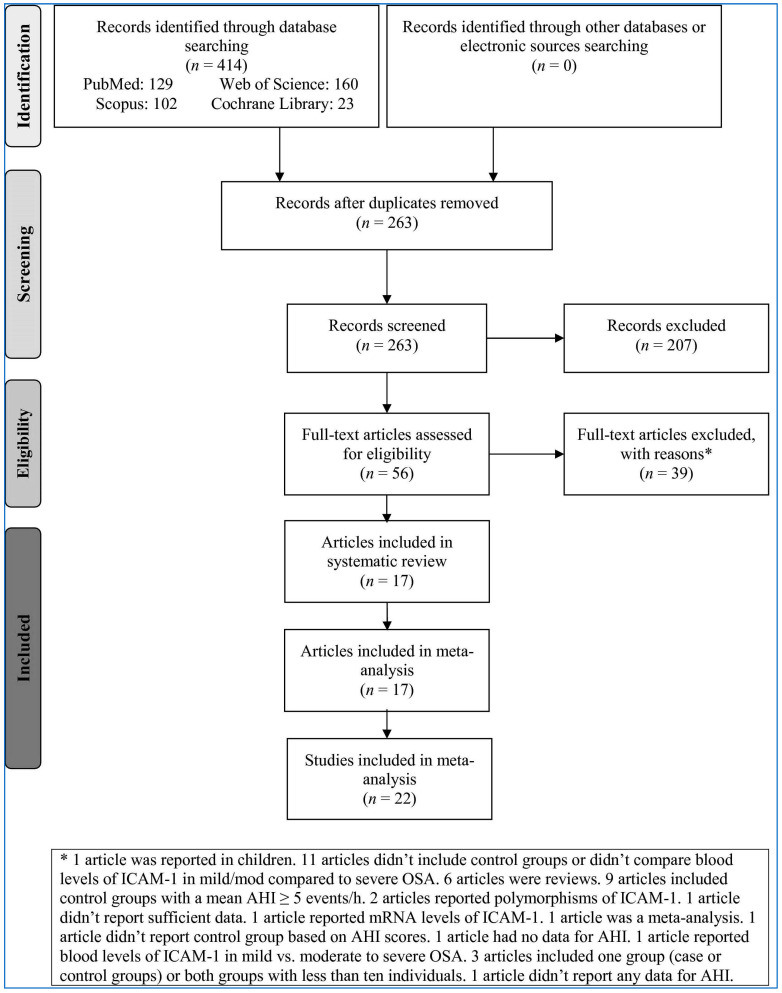
Flowchart of the study selection. ICAM-1: Intercellular adhesion molecule-1. AHI: Apnea-hypopnea index. OSA: Obstructive Sleep Apnea.

**Figure 2 medicina-58-01499-f002:**
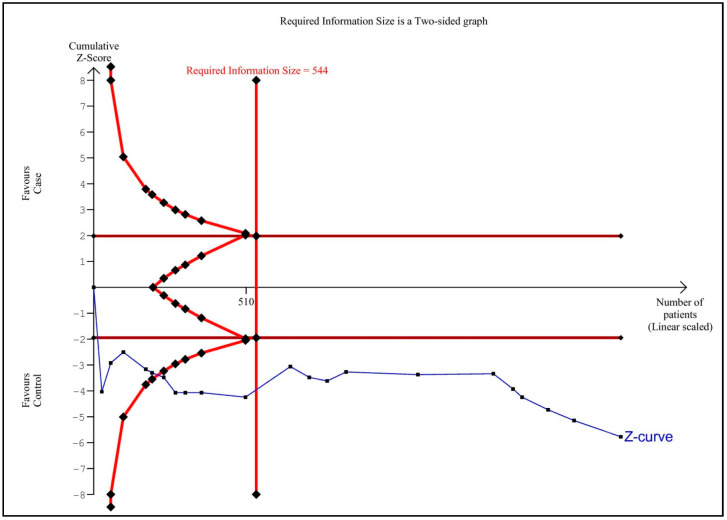
Trial Sequential analysis. Serum/plasma intercellular adhesion molecule-1 levels in adults with obstructive sleep apnea compared to controls.

**Table 1 medicina-58-01499-t001:** Characteristics of articles included in the meta-analysis.

First Author, Publication Year	Ethnicity	Case				Sample	Control				Quality Point
		No.	Mean				No.	Mean			
			Age, Years	BMI, Kg/m^2^	AHI, Events/h			Age, Years	BMI, Kg/m^2^	AHI, Events/h	
El-Solh, 2002 [43]	Mixed	15	61.2	31.47	39.91	Plasma	15	59.3	29.02	3.93	9
Ohga, 2003 [44]	Asian	20	47.8	29.4	38.9	Serum	10	48.9	28.4	3.1	9
Bravo, 2007 [45]	Caucasian	22	52.3	30.9	48.9	Serum	20	47.4	28.4	2.5	8
Ursavaş, 2007 [46]	Caucasian	39	52	30.8	50.5	Serum	34	49	28.8	1.9	8
Carpagnano, 2010 [47]	Caucasian	12	47.3	42.6	48.8	Plasma	10	45.9	34.5	3.9	7
Zamarrón, 2011 [48]	Caucasian	20	50.1	29.9	45.2	Serum	18	44.1	27.6	<5	8
Zhi, 2011 [49]	Asian	20	41.2	28.3	48.8	Serum	20	43.5	26.1	<5	8
Jurado-Gamez, 2012 [50]	Caucasian	68	48	30.54	33.69	Serum	-	-	-	-	8
Chen, 2015 [51]	Asian	20	38.6	27.13	58.89	Serum	14	38.21	23.84	2.86	9
da Silva Araújo, 2015 [31]	Mixed	33	39.6	34.39	20.16	Serum	20	32.5	34.54	2.55	6
Chang, 2017 [52]	Asian	121	43.8	25.1	35.6	Serum	27	39.9	24.4	2.4	9
Jin, 2017 [53]	Asian	100	55.28	26.74	38.01	Plasma	50	56.13	25.19	3.62	9
Xiao, 2017 [30]	Asian	93	70.1	25.31	11.07	Serum	31	68.7	24.92	1.75	8
			68.9	25.12	27.74						
			70.3	29.53	58.83						
Santamaria-Martos, 2018 [29]	Caucasian	228	57.67	28.07	9.33	Serum	132	44.4	24.87	1.89	6
			65.17	28.67	28.6						
Sun, 2019 [54]	Asian	44	43.84	27.59	57.57	Serum	24	44.8	23.42	2.62	7
Nikitidou, 2021 [55]	Caucasian	20	44.2	30.8	48.4	Serum	10	40.2	25.3	3.6	8
Sun, 2022 [56]	Asian	161	47.5	25.13	5–15	Plasma	56	48	23.67	<5	6
			48	26.67	15–30						
			45	29.03	≥30						

AHI: Apnea-Hypopnea Index. BMI: Body Mass Index.

**Table 2 medicina-58-01499-t002:** Results of random-effects forest plot analysis. Serum/plasma intercellular adhesion molecule-1 levels in adults with obstructive sleep apnea compared to controls.

Study, Publication Year	Case			Control			SMD, 95%CI
	Mean	SD	Total	Mean	SD	Total	
Bravo, 2007 [45]	263.0	46.9	22	221.0	38.01	20	0.96 (0.32, 1.60)
Carpagnano, 2010 [47]	100.1	3.6	12	93.3	2.6	10	2.05 (0.98, 3.12)
Chang, 2017 [52]	214.6	78.1	121	183.9	33.0	27	0.42 (0.00, 0.84)
Chen, 2015 [51]	206.93	81.03	20	176.67	35.24	14	0.45 (−0.25, 1.14)
da Silva Araújo, 2015 [31]	105.23	31.19	33	101.38	33.23	20	0.12 (−0.44, 0.67)
El-Solh, 2002 [43]	367.4	85.2	15	252.8	68.4	15	1.44 (0.63, 2.26)
Jin, 2017 [53]	357.92	10.52	100	91.68	53.29	50	8.32 (7.31, 9.33)
Nikitidou, 2021 [55]	471.2	204.5	20	243.6	39.9	10	1.30 (0.47, 2.14)
Ohga, 2003 [44]	448.57	153.79	20	222.14	114.79	10	1.55 (0.68, 2.41)
Santamaria-Martos, 2018 (mild) [29]	148.37	77.8	109	90.55	66.32	132	0.80 (0.54, 1.07)
Santamaria-Martos, 2018 (mod/sev) [29]	88.0	75.68	119	90.55	66.32	132	−0.04 (−0.28, 0.21)
Sun, 2019 [54]	570.17	366.45	44	147.39	185.94	24	1.32 (0.78, 1.87)
Sun, 2022 (mild) [56]	575.6	388.09	29	149.21	255.45	56	1.38 (0.88, 1.87)
Sun, 2022 (mod) [56]	496.02	331.82	33	149.21	255.45	56	1.20 (0.74, 1.67)
Sun, 2022 (sev) [56]	624.6	357.45	99	149.21	255.45	56	1.46 (1.09, 1.82)
Ursavaş, 2007 [46]	480.1	216.7	39	303.4	98.6	34	1.02 (0.53, 1.51)
Xiao, 2017 (mild) [30]	361.7	21.84	31	342.71	17.76	31	0.94 (0.42, 1.47)
Xiao, 2017 (mod) [30]	518.41	30.46	31	342.71	17.76	31	6.96 (5.60, 8.32)
Xiao, 2017 (sev) [30]	711.27	32.67	31	342.71	17.76	31	13.84 (11.27, 16.41)
Zamarrón, 2011 [48]	251.67	69.62	20	221.0	48.15	18	0.50 (−0.15, 1.14)
Zhi, 2011 [49]	118.3	18.3	20	55.3	19.0	20	3.31 (2.33, 4.29)
**Total (95% CI)**			968			797	2.00 (1.41, 2.59)

Heterogeneity: Tau² = 1.70; Chi² = 499.18, df = 20 (*p* < 0.00001); I² = 96%. Test for overall effect: Z = 6.69 (*p* < 0.00001). SMD: Standardized mean difference. CI: Confidence interval. SD: Standard deviation. mod: moderate. sev: severe.

**Table 3 medicina-58-01499-t003:** Results of random-effects forest plot analysis. Serum/plasma intercellular adhesion molecule-1 levels in adults with severe compared to mild/moderate obstructive sleep apnea.

Study, Publication Year	Case			Control			SMD, 95%CI
	Mean	SD	Total	Mean	SD	Total	
Jurado-Gamez, 2012 [50]	435.0	148.89	37	257.33	77.78	31	1.44 (0.90, 1.98)
Sun, 2022 (mild) [56]	624.6	357.45	99	575.6	388.09	29	0.13 (−0.28, 0.55)
Sun, 2022 (mod) [56]	624.6	357.45	99	496.02	331.82	33	0.36 (−0.03, 0.76)
Xiao, 2017 (mild) [30]	711.27	32.67	31	361.7	21.84	31	12.42 (10.11, 14.74)
Xiao, 2017 (mod) [30]	711.27	32.67	31	518.41	30.64	31	6.01 (4.81, 7.21)
**Total (95% CI)**			297			155	3.62 (1.74, 5.51)

Heterogeneity: Tau² = 4.27; Chi² = 188.87, df = 4 (*p* < 0.00001); I² = 98%. Test for overall effect: Z = 3.78 (*p* = 0.0002). SMD: Standardized mean difference. CI: Confidence interval. SD: Standard deviation.

**Table 4 medicina-58-01499-t004:** Subgroup analysis. Intercellular adhesion molecule-1 levels in adults with obstructive sleep apnea compared to controls.

Variable, *N*	SMD	95%CI		*p*-Value	I^2^, %	*P* _heterogeneity_
		Min	Max			
Ethnicity						
Asian (12)	3.07	2.03	4.11	<0.00001	97	<0.00001
Caucasian (7)	0.84	0.38	1.31	0.0004	85	<0.00001
Mixed (2)	0.75	−0.55	2.04	0.26	86	0.009
Sample size						
<100 (16)	1.96	1.32	2.60	<0.00001	93	<0.00001
≥100 (5)	2.07	0.76	3.38	0.002	99	<0.00001
Sampling						
Serum (15)	1.68	1.08	2.28	<0.00001	95	<0.00001
Plasma (6)	2.59	1.17	4.02	0.0004	97	<0.00001

*N*: Number of studies. SMD: Standardized mean difference. CI: Confidence interval.

**Table 5 medicina-58-01499-t005:** Fixed-effects meta-regression analysis. Intercellular adhesion molecule-1 levels in adults with obstructive sleep apnea compared to controls.

Variable	Point Estimate	Standard Error	Lower Limit	Upper Limit	Z-Value	*p*-Value
Publication year	−0.008	0.011	−0.031	0.014	−0.725	0.468
Mean age of adults with OSA	−0.016	0.006	−0.028	−0.004	−2.743	0.006
Mean age of controls	0.081	0.008	0.065	0.097	9.819	<0.001
Mean BMI of adults with OSA	−0.032	0.021	−0.075	0.011	−1.461	0.144
Mean BMI of controls	−0.024	0.021	−0.067	0.018	−1.107	0.268
Mean AHI of adults with OSA	0.016	0.003	0.009	0.024	4.222	<0.001
Sample size	−0.004	0.001	−0.005	−0.003	−7.015	<0.001
Quality point	0.289	0.048	0.194	0.384	5.982	<0.001

OSA: Obstructive Sleep Apnea. AHI: Apnea-Hypopnea Index. BMI: Body Mass Index.

## Data Availability

No new data were created or analyzed in this study. Data sharing is not applicable to this article.

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
