# Peer review of "Evaluation of Blood Intercellular Adhesion Molecule-1 (ICAM-1) Level in Obstructive Sleep Apnea: A Systematic Review and Meta-Analysis"

_medicina, 2022, doi:10.3390/medicina58101499_

Round 1

Reviewer 1 Report

lines 28-30. Explain in the abstract why “in addition to the meta-analysis, the following analyses were carried out: Sensitivity analysis, subgroup analysis, trial sequential analysis, meta-regression, and a funnel plot analysis explain why were the extra analysis needed?

Line 70. The connection between ICAM-1 and the inflammatory response should be expanded and better explained

Lines 94-97. This question should be presented also in the abstract/introduction

141 Statistical analysis. Can the statistical analysis be condensed?

182-192 Unnecessary information. It can be added to the supplementary data section (if at all)

I wonder if Figures 2,3,4,5 should be labeled as tables.

In sections 3.3 and 3.4 there should be a clearer description of the results and their significance.

English should be improved. Examples: What is meant by 48- ×´define OSA×´, diagnose? characterize? ; 51-Higher age?; correct 87- “Trtrial”

49-51- It should be better explained, mostly for the non-OSA expert reader.

Should the color of the figures be changed? (not brown/orange)

Resolution and clarity of the figures 2,3,4,5 should be improved

Each figure/table should have a title. Don’t start the explanation of the results as: “Figure 2 illustrates…” or “Figure 3 shows…”

The legend in figure 2 should be attached to the bottom of the figure and not in a new page

259-260- what is the difference between “heterogeneous” and “high heterogenicity”, what is the significance of the difference?

319- why “in contrast” may be it should be “in addition”?

Author Response

We thank Reviewer #1 for their valuable and encouraging comments, which helped us to improve the quality of the manuscript. Please find the revised manuscript and the detailed point-by-point-response attached as separate files. Again, we thank Reviewer #1 for the care devoted to the present manuscript. 

Reviewer 2 Report

The present systematic review and meta-analysis assessed the ICAM-1 levels in serum and plasma in people with Obstructive Sleep Apnea.

I would like to congratulate the authors for the thorough statistical analysis.

The review is generally well conducted and written, but there is room for some improvements.

Major issues

Methods

Line 138: As stated in Cochrane Collaboration Handbook, the use of scores for instruments assessing the risk of bias is highly discouraged! Thus this score should be removed. Moreover, a Table with the NOS evaluation per each study and detailed presentation in text of the synthesis of the situation for each domain of NOS results, as well an adequate discussion per domain of bias is warranted, and finally it should be added to the limitations subchapter.

Minor issues

Methods

You stated ”The design of present meta-analysis is in accordance with the Preferred Reporting

Items for Systematic Reviews and Meta-Analyses (PRISMA) protocols”, but PRISMA is a reporting guideline, not a guideline to design and conduct a systematic review, like Cochrane Handbook, for example. Thus please rewrite the sentence similar to: ”We reported the systematic review according to the PRISMA guideline”

Line 103: Report how the age and sample size restrictions were performed for each database separately in a supplementary file (eg. using filters, or by manual selection). In case automated measures were used (i.e. filters) then please add this as a limitation in Discussion, since many articles are not indexed in databases with this information (for example those in PubMed Central).

Kindly provide the full strategy for Pubmed or EMBASE (as stated in PRISMA), in a supplementary file.

Line 118: 4) Studies with 10 or more – this is an exclusion criteria, it should be removed to avoid redundancy.

Line 127: 4) including pediatric samples aged < 18 years – it is the negation of an inclusion criteria, it should be removed to avoid redundancy.

Line 131: “imported in the meta-analysis” – more likely selected for

Line 173: In another way, the volume of in-

formation was not wide enough and more individuals were needed. - kindly reformulate

Results

Was the meta-regression univariate or multivariate?

A meta-regression for a methodological quality of the study would be interesting

The statistical exercise on heterogeneity assessment lacks any clinical touch. What are the clinical characteristics of the outlier studies?

Kindly move less important charts to a Supplementary file: sensitivity analyses, the Galbraith (or Radial) and L’Abbe plots, and the funnel plot, and table 4, they put in shadow the main results.

Author Response

We thank Reviewer #2 for their valuable and encouraging comments, which helped us to improve the quality of the manuscript. Please find the revised manuscript and the detailed point-by-point-response attached as separate files. Again, we thank Reviewer #2 for the care devoted to the present manuscript.

Round 2

Author Response

Again, we thank Reviewer #2 for their valuable comments and suggestions. Please find attached the detailed point-by-point-response. 

Again, we appreciated the support to improve the quality of the present revision.

Round 3

Reviewer 2 Report

Thank you for the answers and for all the work! No further questions.